# Organic Electrochemical Transistors as Versatile Tool for Real-Time and Automatized Viral Cytopathic Effect Evaluation

**DOI:** 10.3390/v14061155

**Published:** 2022-05-26

**Authors:** Francesco Decataldo, Catia Giovannini, Laura Grumiro, Maria Michela Marino, Francesca Faccin, Martina Brandolini, Giorgio Dirani, Francesca Taddei, Davide Lelli, Marta Tessarolo, Maria Calienni, Carla Cacciotto, Alessandra Mistral De Pascali, Antonio Lavazza, Beatrice Fraboni, Vittorio Sambri, Alessandra Scagliarini

**Affiliations:** 1Department of Physics and Astronomy, Alma Mater Studiorum, University of Bologna, 40127 Bologna, Italy; francesco.decataldo2@unibo.it (F.D.); marta.tessarolo3@unibo.it (M.T.); maria.calienni@unibo.it (M.C.); 2Department of Experimental, Diagnostic and Specialty Medicine—DIMES, Universtity of Bologna, 40138 Bologna, Italy; catia.giovannini4@unibo.it (C.G.); carla.cacciotto@unibo.it (C.C.); alessandra.depascal3@unibo.it (A.M.D.P.); alessand.scagliarini@unibo.it (A.S.); 3Center for Applied Biomedical Research (CRBA), S. Orsola-Malpighi University Hospital, 40138 Bologna, Italy; 4Unit of Microbiology, The Great Romagna Hub Laboratory, 47522 Pievesestina, Italy; laura.grumiro@auslromagna.it (L.G.); mariamichela.marino@auslromagna.it (M.M.M.); martina.brandolini@studio.unibo.it (M.B.); giorgio.dirani@auslromagna.it (G.D.); fra.taddei@hotmail.it (F.T.); 5Experimental Zooprofilactic Institute of Lombardy and Emilia Romagna“Bruno Ubertini” (IZSLER), 25124 Brescia, Italy; francesca.faccin@izsler.it (F.F.); davide.lelli@izsler.it (D.L.); antonio.lavazza@izsler.it (A.L.)

**Keywords:** organic electrochemical transistor, bovine coronavirus, BCoV, encephalomyocarditis virus, ECMV, cytolytic virus, non-cytolytic virus, virus replication

## Abstract

In-vitro viral studies are still fundamental for biomedical research since studying the virus kinetics on cells is crucial for the determination of the biological properties of viruses and for screening the inhibitors of infections. Moreover, testing potential viral contaminants is often mandatory for safety evaluation. Nowadays, viral cytopathic effects are mainly evaluated through end-point assays requiring dye-staining combined with optical evaluation. Recently, optical-based automatized equipment has been marketed, aimed at the real-time screening of cell-layer status and obtaining further insights, which are unavailable with end-point assays. However, these technologies present two huge limitations, namely, high costs and the possibility to study only cytopathic viruses, whose effects lead to plaque formation and layer disruption. Here, we employed poly(3,4-ethylenedioxythiophene):poly(styrene sulfonate) (Pedot:Pss) organic electrochemical transistors (OECTs) for the real-time, electrical monitoring of the infection of cytolytic viruses, i.e., encephalomyocarditis virus (EMCV), and non-cytolytic viruses, i.e., bovine coronavirus (B-CoV), on cells. OECT data on EMCV were validated using a commercially-available optical-based technology, which, however, failed in the B-CoV titration analysis, as expected. The OECTs proved to be reliable, fast, and versatile devices for viral infection monitoring, which could be scaled up at low cost, reducing the operator workload and speeding up in-vitro assays in the biomedical research field.

## 1. Introduction

In vitro cell culture studies are key systems required for virus cultivation and for many research purposes, such as investigating virus–cell interactions and screening potential inhibitors of infections, such as antibodies and antiviral drugs. The importance of cell cultures to study viral replication and to perform a variety of laboratory tests is evidenced by the high number of scientific papers that are still presenting results obtained using the cell culture system. It is worth remembering that cell-based infectivity assays are crucial in several areas: virus production allows for the manufacturing of different vaccines, and virus–cell interactions are investigated for virotherapy (representing potential treatments in immunotherapy and gene therapy, and to tackle several human diseases, including cancer) [1,2,3]. Moreover, drug safety evaluation often requires the testing of potential viral contaminants (as recommended by the Food and Drug Administration, FDA) [4]. At present, the determination of the biological properties of new viruses detected via metagenomics is posing a colossal challenge to assess their taxonomy as this is still relying on phenotypic properties such as morphology, genome type, and replication in cells [5].

Various pathological cell alterations, triggered by eukaryotic viruses, are referred to as the cytopathic effect (CPE) [6]. Viral CPEs on cells are investigated by means of dye-based, labor-intensive, end-point assays or imaging. New optical instrumentations have recently been commercialized, which can be useful to observe the real-time effect of viral infection and replication on cell-layer status. These technologies are very expensive, limiting their spread and utilization to a small number of laboratories/institutes, and few data, regarding their ability to detect the effects of viral infection on cell lines, are currently available. Thus, high-throughput, low-cost, and scalable systems would greatly help researchers, diagnosticians, and pharmaceutical and biomedical companies to reduce development and test costs while speeding up experimental processes and studies. From this perspective, products based on alternating-current (AC), electrical measurements, such as xCELLigence and Maestro Z, have been developed to evaluate cell viability, proliferation, cytotoxicity, bacterial toxins, and viral infection or titration, by measuring cell-layer impedance [7,8,9,10,11,12,13,14]. These systems allow for the real-time assessment of the cell culture under study, providing further insights compared to standard end-point assays, with the aim of substituting or integrating optical imaging. Accordingly, in our previous work, we demonstrated that the planar Pedot:Pss-based OECTs could be used for the fast and real-time electrical detection of the serum-neutralization assay, for quantifying SARS-CoV-2 neutralizing antibodies serum titer [15]. Owing to its aspecific nature, the proposed technology should not be limited to SARS-CoV-2 virus since the OECTs can be used to monitor other virus-induced CPEs during in vitro replication. With this purpose, we monitored the in vitro replication of the encephalomyocarditis virus (EMCV) and bovine coronavirus (B-CoV). EMCV is a small, non-enveloped virus with a positive single-strand RNA genome (7.8 kb) belonging to the cardiovirus genus in the *Picornaviridae* family. EMCV has worldwide distribution, it can infect a wide range of animal species and is also a potential zoonotic agent. The EMCV has mainly been described as a lytic virus causing “necrotic” cell death [16]. B-CoVs are enveloped pleomorphic pneumo-enteric viruses with a positive single-strand RNA genome (26.4–31.7 kb) that belong to the genus betacoronavirus, a subfamily of *Coronavirinae* within the family *Coronaviridae,* order Nidovirales, associated with neonatal calf diarrhea and with winter dysentery and shipping fever in older cattle [17]. B-CoVs are capable of interspecies transmission and causing disease in adoptive/spillover hosts. Besides bovine cell substrates [18], BCoVs can readily replicate in human rectal tumor-18 (HRT-18) cells, suggesting a possibility for zoonotic transmission events [17]. B-CoV’s observed CPE is a progressive rounding and loss of adherence of cells associated with the granularity and formation of syncytia [18].

Here, we introduced the planar Pedot:Pss-based OECTs for monitoring the effects of the viral replication of these two viruses causing different CPEs in their respective permissive cell substrates. The results were compared to an optical automatized instrumentation used as a gold standard technique.

## 2. Materials and Methods

### 2.1. OECT Device Fabrication

Glass substrates (25 × 25 mm^2^) were cleaned by sonication in distilled water/acetone/isopropanol baths. Afterwards, 10 nm of chrome and 50 nm of gold were deposited by thermal evaporation. Substrates were then treated with air plasma (20 W for 4 min), and then the PEDOT:PSS solution was spin-coated (3000 rpm for 10 s) using a Teflon mask. The thin film thickness was 140 ± 10 nm. The solution was made of 94% PEDOT: PSS (Clevios PH1000, provided by Heraeus Deutschland GmbH & Co., Leverkusen, Germany) with 5% of ethylene glycol (EG) (Sigma-Aldrich, St. Louis, MO, USA), 1% of 3-glycidoxypropyltrimethoxysilane (GOPS) (Sigma Aldrich, St. Louis, MO, USA), and 0.25% of 4-dodecylbenzenesulfonicacid (DBSA) (Sigma Aldrich, St. Louis, MO, USA). This suspension was treated in an ultrasonic bath for 10 min and filtered using 1.2 μm cellulose acetate filters (Sartorius) before the deposition. The samples were subsequently baked at 120 °C for 1 h. The planar geometry OECTs were patterned, having two channels with a length (*L*) of 1 mm, a width (*W*) of 0.75 mm, and an inner gate electrode (*L* = 2 mm, *W* = 3 mm). Then, the devices were merged in distilled H_2_O for 1 h and dried with a nitrogen flux. In the end, a polydimethylsiloxane (PDMS) transparent, cylindrical well, having an inner diameter and a height of 12 mm and 8 mm, respectively, was bound to the device to realize the culture well.

### 2.2. OECT Integrated System and Electrical Characterization Set-Up

All measurements were performed in an MEM electrolyte solution. Experiments were performed using an integrated system, the TECH-OECTs, reported in Figure 1 and in our previous work [19]. Cells were seeded inside cylindrical PDMS wells, having a diameter of 12 mm, and measurements were started 1 h after seeding for all the reported experiments. A multiplexer system was used to sequentially measure 12 channels. We measured the source-drain current by means of a Keysight B2912A Source Measure Unit, while biasing the channel with V_ds_ = –0.1 V and introducing a square wave potential on the gate electrode, from V_gs(OFF)_ = 0.0 V to V_gs(ON)_ = 0.3 V, with *t*_on_ = 0.5 s and *t*_off_ = 1.5 s. The keysight and the multiplexer were both controlled with a customized PC software.

### 2.3. OECT Data Analysis

Output data were analyzed with a customized Matlab routine: each single channel response to a pulse on the gate was isolated, normalized, and fitted with the bi-exponential curve $I_{d}=a\;exp^{-\frac{x}{\tau_{1}}}+exp^{-\frac{x}{\tau_{2}}}+e$, as reported in our previous work [19]. As described elsewhere [20], labelling $\tau_{1}$ greater than $\tau_{2}$, $\tau_{1}$ represents the charging time of the PEDOT:PSS influenced by the ion-blocking properties of the cell layer, while $\tau_{2}$ relates to the charging time of the cell layer. Thus, we will focus and examine $\tau_{1}$ as the device time response to a gate potential pulse, averaging its value over five pulses on the same channel and then normalizing, using the following equation: T time response (a.u.)=ττNo Cells, with $\tau_{No\;Cells}$ representing the response time of the device before cell seeding. Data are shown in arbitrary units (a.u.) after the normalization, with the standard deviation of each point reported as the curve shadow.

### 2.4. Viruses and Cell Lines

The viruses used in this study were the encephalomyocarditis virus (EMCV) field strain IZSLER 425006/2019 isolated from pigs and the bovine coronavirus (B-CoV) strain 9WBL77. VERO E6 (ATCC CRL-1586) and HRT-18 (BS TCL26, Biobanking of Veterinary Resources, IZSLER)) were used for experiments with EMCV and B-CoV, respectively. Cells were maintained in MEM supplemented with 10% heat-inactivated fetal bovine serum (FBS), 1% penicillin–streptomycin (P/S), and 1% l-glutamine (l-Gln). Culture medium and supplements were all purchased from EuroClone (Milan, Italy).

### 2.5. Cell Plating and Infection

Two infection protocols were used, defined as “adhesion” and “suspension” throughout the manuscripts. For the adhesion method, HRT18 and VERO E6 were seeded in 24-well plates or on the OECTs at a density of 30,000 and 15,000 cells/well, respectively. After reaching confluence, the medium was aspirated and the wells were washed with fresh full medium supplemented with EMCV (TCID50 = 10^6^/50 µL) or B-CoV (TCID50 = 10^5.8^/50 µL) at different dilutions. For the suspension method, HRT18 and VERO E6 cells were mixed with EMCV or B-CoV at different dilutions, seeded in 24-well plates or on the OECTs, and allowed to fall onto the device before starting the measurements.

The cells were monitored by using the Incucyte^®^ live-cell imaging system (Essen BioScience, Royston, UK) or the OECT devices. In Incucyte^®^, cells were photographed automatically at the same place and at specific time intervals. This imaging system automatically calculates the cell area and cell density percentage at each time point. The same experiments were repeated in suspension by mixing viruses with cells before seeding them in 24-well plates or on the OECTs. The differences between the controls and the infected cells were analyzed using a double-sided Student’s *t*-test. *p*-values < 0.05 were considered statistically significant. Statistical analyses were performed using SPSS version 19.0 (IBMCorp., Armonk, NY, USA).

### 2.6. Cell Viability Evaluation

Cell death was detected by measuring the permeability of the plasma membrane to the normally impermeable fluorescent dye propidium iodide (PI). At the end of the experiment, cells were incubated with PI for 10 min at room temperature. Optical micrographs were taken using an inverted optical microscope, with magnification 20× (EVOS, ThermoFisher, Waltham, MA, USA). Cells were then harvested, and PI uptake was quantified by FACS analysis (Cytoflex-S Beckman Coulter, Brea, CA, USA).

## 3. Results

Planar, Pedot:Pss-based organic electrochemical transistors were fabricated as described in Materials and Methods section, with the same geometry used in our previous study on SARS-CoV-2 serum-neutralization [15]. The schematic of the device chip for the single culture evaluation is reported in Figure 1a, while Figure 1b shows the electronic board, named TECH-OECT, which allows for multiplexed measurements of up to six cell cultures inside the incubator.

In this work, the OECTs were employed to assess the viral infection on cells of a cytolytic and a non-cytolytic virus, namely, encephalomyocarditis virus (EMCV) and bovine coronavirus (B-CoV), respectively. Contrary to non-cytolytic, the infection by cytolytic viruses results in the disruption of the cell membrane; shrinkage; and the consequent detachment of the cell from the culturing substrate.

A schematic and an optical micrograph of the described viral activities are reported in Figure 1c,d for EMCV and B-CoV, respectively.

### 3.1. B-CoV

We used the HRT-18 cell line since this represents a permissive substrate for B-CoV and is widely employed to monitor virus replication [18,21]. Appendix A reports B-CoV effects when incubated with HRT-18, in comparison to a standard healthy cell monolayer over time. Cells were directly grown on top of the OECT device (seeded as a standard petri well, as visible in Figure 1a), having optically transparent and biocompatible Pedot:Pss thin films as their active areas. Vacuoles formation starts 24 h after the virus addition to receive a boost after 48 h, hiding the cell-layer view and thus hindering the optical evaluation of the cell health with a standard automatized system, such as the optical microscope Incucyte^®^. The Pedot:Pss-based planar OECTs allowed for the real-time, electrical monitoring of the B-CoV viral infection on HRT-18, which was infected in suspension and adhesion; for OECT devices, the analysis of the cell-layer integrity relies on the normalized current modulation induced by a voltage potential applied on the gate terminal. Appendix A clearly shows that the device response is slowed down by the cells integrated on the chip (green and blue filled triangles); on the contrary, B-CoV infection shifts the curve back towards faster time responses (orange and red empty triangles), close to the current modulation obtained when no cells are present on top of the device (bare OECT, and black, empty circles). Accordingly, the extraction of the OECT time response parameter allows for the real-time monitoring of cell-layer health upon exposure to this non-cytolytic viral agent for 72 h, as shown in Figure 2a,b for the suspension and adhesion protocol, respectively. It is worth noting that virus-infected cells immediately showed lower OECT time responses (i.e., faster ion flux due to the viral-induced reduction in the barrier properties of the cell layer), but a visible cytopathic effect occurs only 48 h after virus incubation [22]. This outcome is well-correlated with the strong appearance of numerous vacuoles in the optical micrographs reported in the Supporting Information (Appendix A). Furthermore, the stable time response of the device over 72 h, when incubated only with B-CoV without cells seeded (Appendix A), proves that the OECT signal is not affected by the viral agent. Thus, the results in Figure 2a,b represent the stress suffered by the cells and caused by the B-CoV infection. Fluctuations that may be present in the curves are likely related to transitory biological events, cell culture inhomogeneity, and/or external electrical noises/mechanical vibrations; moreover, moving the TECH-OECT prototype for taking optical images every 24 h might move cells not adhered/detatched to/from the surface, slightly imparting on the device response. However, a complete trend analysis and real-time acquisition allow for the extraction of the correct assay outcome, which is unaffected by likely signal fluctuations. Finally, it has to be noted that viral infection in non-adherent cell culture produces better results, displaying a stronger discrimination between the healthy and infected cells. Therefore, further measurements with B-CoV were carried out with the suspension protocol, also owing to the minor time required for the experimental outcome.

Figure 2c,d report Incucyte^®^ data for HRT-18 cells grown on a standard petri multi-well, infected with B-CoV in suspension and adhesion, respectively. It is worth noting that no differences are present between a viral-infected (red dotted line) and a healthy cell culture (CTRL, black squared line). Indeed, Incucyte^®^ technology relies on optical masks locating cell-layer presence or absence (empty spaces) and fails in detecting the viral effect since the non-cytopathic virus does not lead to cell-layer plaque or disruption and the vacuoles formation prevents cell-layer focusing. The cytopathic effect that lowers OECT time response after 48 h from the virus incubation induced us to stop the Incucyte^®^ experiments in adhesion (Figure 2d) two days post infection when no differences were still appreciated in cell growth between infected and healthy cell cultures. Appendix A reports the images acquired by this optical-based technology at the end of the experiment, confirming the plotted data.

Noting the large difference in the OECT time response when monitoring B-CoV-infected and healthy HRT-18 growth in suspension, we investigated the use of our technology for testing viral dilution effects. We started from the concentrated virus used for the previous experiments (TCID50 = 10^5.8^/50 µL), realized five progressive dilutions, and then incubated with HRT-18 cell lines for 72 h. Figure 3a reports the OECT time response for this experiment, proving that the devices correctly discriminate the progressively milder effects induced by the various viral dilutions. On the other hand, the optical-based technology Incucyte^®^ struggled to appreciate these differences (Figure 3b), as expected on the basis of the previous experiment. Noteworthily, the cytofluorimeter analysis (Figure 3c) and the optical micrographs (Appendix A, red staining with Evos to show dead cells), acquired as end-point assays after 72 h, confirmed our electrical results, both reporting the progressive reduction in the cell dead population upon viral dilution. Noteworthily, all the dilutions produced a cytopathic effect on HRT-18 since the maximum normalized time response (1.30 a.u.) is strongly below the usual values for healthy cell growths (in close proximity to 2 a.u.).

### 3.2. EMCV

We proved the versatility of our technology for in-vitro, viral infection studies using the OECTs and using a cytolytic virus encephalomyocarditis virus (EMCV), grown on the VERO E6 cell line. It is well established that EMCV causes detrimental CPEs on the cell monolayer, leading to plaque formation and cell-layer destruction (as reported in Appendix A), and optical techniques can be used to evaluate cell status. Thus, we compared and validated the electrical-based OECT response signal with the commercially-available, optical-based Incucyte^®^ device. Again, we monitored EMCV activity in adhesion and suspension. Figure 4 reports the EMCV effect on VERO E6 cells upon different dilutions of viral inoculum, using the adhesion infection protocol. Cells were cultivated directly on the OECTs and on standard multi-wells substrates, introducing EMCV after the formation of a first-cell monolayer (i.e., 24 h post seeding). Then, the virus infection was monitored for 48 h electrically (Figure 4a, data are normalized right after the seeding) and optically (Figure 4b, data are normalized after the viral infection). The signal from the untreated cells (black dotted line) was used as a control, representing the uninfected cells’ standard growth curve. OECT data clearly show mortality for all viral dilutions, but higher final time responses are correlated with more diluted viral concentrations. It is worth noting that 24 h after seeding, a signal spike is present almost for all cell cultures, being more evident for the control and viral diluted ones. We hypothesized that taking optical images on TECH-OECT and medium additions/substitutions could modify the non-adhered cell position, thus changing the cell-layer overall resistance. However, the OECTs correctly detect the different viral dilutions, thus proving that real-time analysis overcomes signal fluctuations derived from operator interactions with the cell culture. On the contrary, Incucyte^®^ technology only confirms the stress induced by the highest viral concentration, while it is unable to differentiate between the effects of the other viral dilutions and the uninfected control. Optical micrographs taken on Pedot:Pss at the end of the experiment, red-stained with Evos to highlight the dead cells, confirmed a stronger CPE effect at higher viral concentrations. Similarly, cytofluorimeter analysis detected a higher percentage of dead cells for the more concentrated viral inocula (Appendix A). The cell layers evaluated on OECT 48 h post infection were strongly damaged for lower viral dilution (1:30; 1:100), suggesting that the effects of these viral dilutions can be detected by Incucyte^®^ lengthening of the infection’s time. However, the optical images taken on Pedot:Pss showed an intact cell monolayer for 1:300 viral dilution, even if the cells caused a lowering contrast compared to the control cells (Figure 4c). This is probably due to a cellular suffering status that could be detected only by OECT and not by Incucyte^®,^ which displayed an intact cell layer. A constant time response was detected when the OECT was incubated with EMCV virus only (Appendix A), indicating that the viral strain does not alter the device performances, and the obtained results mirror the cell-layer status. It is worth highlighting that the OECTs also detected the EMCV effect on the cells at the extreme dilution (1:3000), while the Incucyte^®^ technology struggled to distinguish the stronger viral dilution from the not infected control.

The same experiment has been replicated culturing the VERO E6 cells and infecting the culture in suspension. In this protocol, cells were incubated with the EMCV virus right after seeding, to speed up the assay. Data are reported in Figure 5: the OECTs (Figure 5a) and Incucyte^®^ (Figure 5b) strongly discriminate the deep drop of the response time of the cultures infected with the highest concentration of EMCV, when compared to the constant growth of the non-infected control. Moreover, progressively higher trends were detected for increasing dilutions, which are slightly more defined for Incucyte^®^. In particular, both technologies reported a high increase between the highest virus concentration (red dots) and the second-highest one (blue dots), with reducing gaps for the gradual dilutions. Again, the optical images with red dye staining dead cells (Figure 5c) are taken at the end of the experiment for confirming the obtained results. Finally, the OECTs provided earlier detection of the viral infected cells, in comparison to the healthy ones.

## 4. Discussion

To date, the “gold standard” techniques for investigating and monitoring the in vitro viral-induced cytopathic effect (CPE) consist of the direct observation of the virus-induced morphological changes and the damage of the cells after a variable period, depending on the viral agent. These methods are end-point assays that do not provide rapid turnaround to results; it takes between 48 h and several weeks for the development of CPE to become apparent [23]. Furthermore, non-cytolytic viruses induce a plethora of different CPEs, such as cell rounding, the appearance of nuclear or cytoplasmic inclusion bodies, and the formation of syncytia that can be difficult to interpret and may require dye-based techniques. Thus, all these methodologies require high-cost and cumbersome optical-based equipment and highly specialized personnel and laboratories.

To develop an alternative strategy for monitoring the cell damage/death, we introduced planar, the Pedot:Pss-based organic electrochemical transistors (OECTs), to assess and monitor cell-layer health in real-time and with fast temporal resolution. The device’s time-response velocity upon a positive voltage pulse on the gate (i.e., the time required for current switching from the “on” state to the “off” one) can be directly correlated to the tissue integrity and health of the cell layer, which is directly grown onto the OECT [20,24]. Thus, our system is potentially versatile towards different cell lines and viruses. The devices were embedded in a prototype-integrated system, called the tissue engineering cell holder for the organic electrochemical transistors (TECH-OECTs), presented in our previous works; the OECTs were employed to investigate cell-layer formation and detachment [19] and the cytotoxicity induced by toxic external agents [25], and for the SARS-CoV-2 serum-neutralization assays [15]. In particular, the latter work demonstrated the OECT capability to promptly evaluate SARS-CoV-2 effects on VERO E6 cells, proving the reliability and reproducibility of the sensing devices, together with their re-usability for up to three consecutive neutralization assays.

In this study, we extended our technique towards the evaluation of viral effects induced by both cytolytic and non-cytolytic viruses, using EMCV and B-CoV as representatives of the two classes. Measurements were compared with the optical-based Incucyte^®^ technique, which was employed as the gold standard. Noteworthily, we have demonstrated the ability of our devices to monitor two different CPEs resulting in cell lysis and cell degeneration (vacuoles formation), induced, respectively, by EMCV on VERO E6 and B-CoV on HRT-18 cell lines. The virus effects on cell cultures have been assessed both for suspension and adhesion-infection protocols. Viral dilutions were also tested, obtaining progressively higher OECT time responses upon decreasing viral concentration, highlighting a reduction in cell dead population, as confirmed by standard cytofluorimetric assays and optical evaluations. It is worth noting that OECT technology succeeded in correctly monitoring the B-CoV non-cytolytic effect on HRT-18 cells: despite virus mild influence on cells, the OECTs are sensitive enough to appreciate the stress induced on HRT-18, decreasing the device time response owing to the reduction in cell tissue resistance towards ion flow caused by internal cell changes (such as vacuolization). The time response increment after viral infection may be similarly explained, as previously reported [18], by cell degeneration (swelling, size changes, vacuoles, and syncytia formation). On the contrary, the optical-based Incucyte^®^ technology failed in quantifying B-CoV action on the cells since optical masks lack the precision to outline a healthy cell from a stressed one, especially when vacuole formation hinders their visual/focus. Moreover, it was observed that the OECTs provided faster discrimination between the highly concentrated EMCV-infected cells and the controls. EMCV dilutions using the adhesion protocol were clearly discriminated by the OECTs, while Incucyte^®^ technology was able to identify only the highest viral concentration; on the other hand, slightly better responses were reported by the Incucyte^®^ for EMCV-diluted cultures when infected in suspension.

We proved that our technology is reliable and effective for monitoring the replication of the two viral species causing different types of CPE in their respective cell substrate, comparing our sensor outputs with the live-cell imaging provided by a commercially available analysis system.

In conclusion, the OECTs represent a new economic approach to ease and reduce the costs of viral infection/proliferation/titration studies. Their versatile nature spreads the possibility to investigate several cell–virus binomials, capturing, in real-time, the cytopathic effect on the culture. The scalability of the low-cost devices and the electronic readout, together with the automatized data extraction and analysis, could decrease scientists’ workloads and speed up in-vitro tests on several viral pathogens.

## 5. Patents

There is a patent pending on this technology, regarding its use for viral serum-neutralization assays.

## Figures and Tables

**Figure 1 viruses-14-01155-f001:**
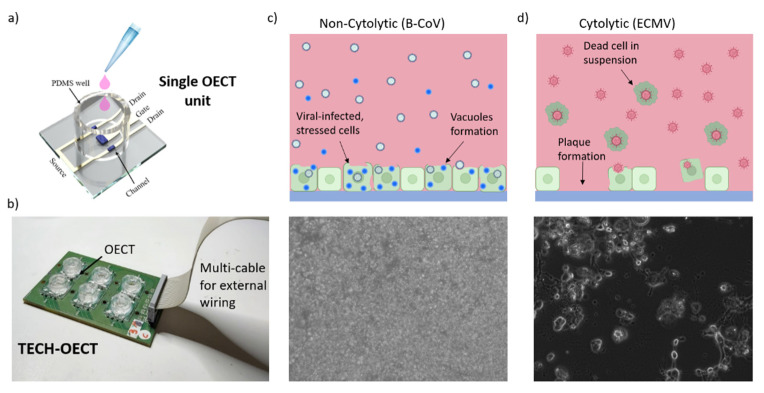
(**a**) An OECT render schematic. (**b**) The TECH-OECT system used for the measurements inside the incubator. The schematic non-cytolytic (**c**) and cytolytic (**d**) viral action (top) and micrograph (bottom) are on the cell lines.

**Figure 2 viruses-14-01155-f002:**
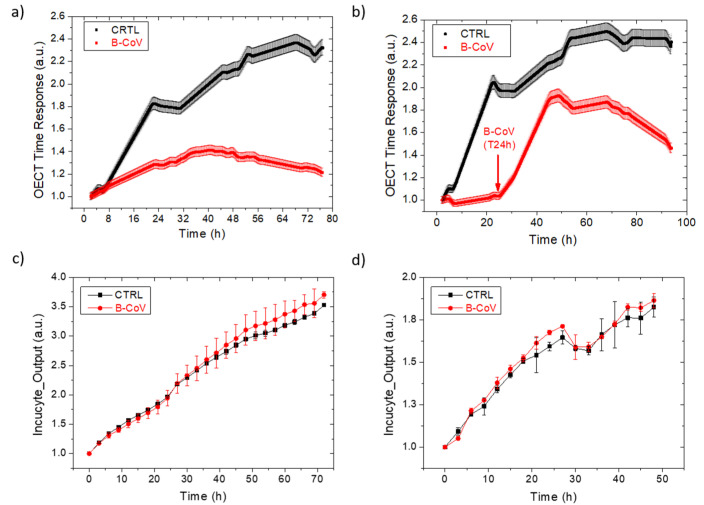
The OECT real-time monitoring of HRT-18 cell growth infected with B-CoV in suspension (**a**) and adhesion (**b**) for 48 h. Each OECT time response point is the average of five consecutive measurements, having the standard deviation reported as the curve shadow. The normalized Incucyte^®^ data of HRT-18 cell growth, infected with B-CoV in suspension (**c**) and adhesion ((**d**), the data were taken and normalized after viral incubation). The experiments were stopped 72 h post virus infection, except for the Incucyte screening in adhesion, which was stopped after 48 h since a clear cytopathic effect had already been monitored by the OECTs.

**Figure 3 viruses-14-01155-f003:**
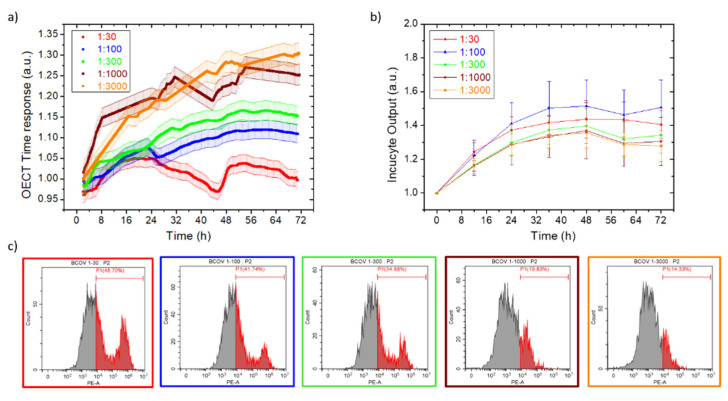
OECT (**a**) and Incucyte^®^ (**b**) real-time monitoring of HRT-18 cells, infected with progressively diluted B-CoV concentrations in suspension. Each OECT time response point is the average of five consecutive measurements, having the standard deviation reported as a curve shadow. (**c**) The cytofluorimeter analysis of the HRT-18 dead population after 72 h of B-CoV incubation, at increasing viral dilutions (from left to right).

**Figure 4 viruses-14-01155-f004:**
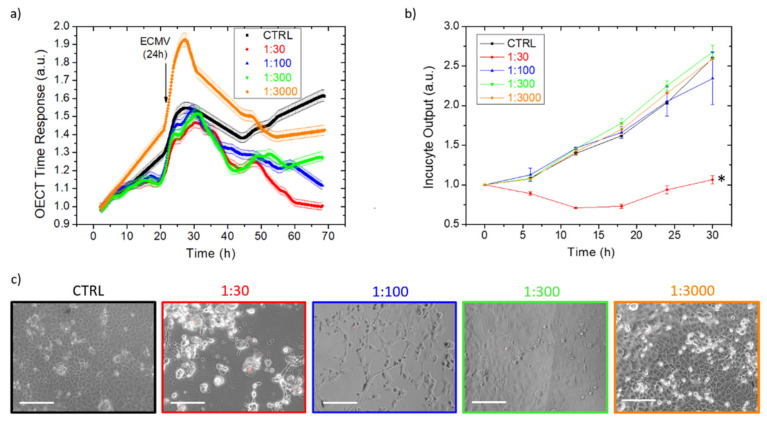
The OECT (**a**) and Incucyte^®^ (**b**) real-time monitoring of VERO E6 cells, infected with progressively diluted EMCV inocula, compared to a healthy cell growth, as the control. Each OECT time response point is the average of five consecutive measurements, having the standard deviation reported as the curve shadow. (**c**) The optical micrographs on the active area of the devices after 48 h of incubation, with different dilutions of EMCV and the control over a standard healthy growth, using a red staining dye for dead cells. White scale bars = 150 µm. * *p* < 0.05 denotes a significant difference compared to the control.

**Figure 5 viruses-14-01155-f005:**
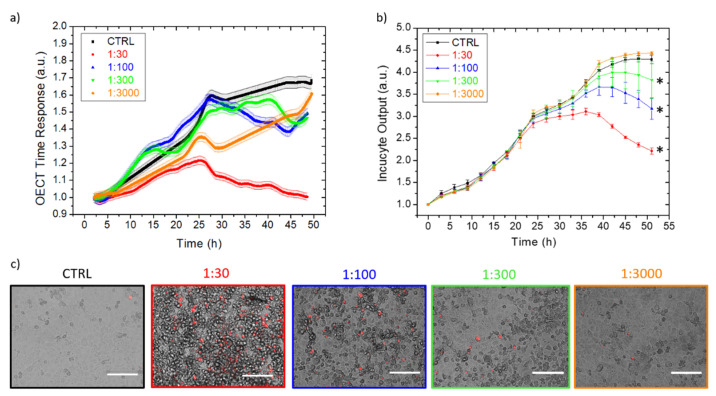
The OECT (**a**) and Incucyte^®^ (**b**) real-time monitoring of VERO E6 cells, infected with progressively diluted EMCV, compared to healthy cell growth, as the control. Each OECT time response point is the average of five consecutive measurements, having the standard deviation reported as the curve shadow. (**c**) The optical micrographs on the active area of the devices after 48 h of EMCV incubation at different dilutions and the control over a standard healthy growth, using a red staining dye for dead cells. * *p* < 0.05 denotes a significant difference compared to the control.

## Data Availability

Data are available on reasonable request from the corresponding authours or the first author, due to the presence of a patent pending on the reported technology.

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
