# Peer review of "Organic Electrochemical Transistors as Versatile Tool for Real-Time and Automatized Viral Cytopathic Effect Evaluation"

_viruses, 2022, doi:10.3390/v14061155_

Round 1

Reviewer 1 Report

The quality of pictures in figure has to be improved,especially,Figure 2a ,2b,Figure 3a,Figure 4a and Figure 5a.It is too blurry.

In figure 3a,the 1:30 and 1:1000 group both decline after about 48 hours and rises up while other groups not.How do you explain this result.

In figure 4a, the control group decline after about 24hours then goes up.It should be constantly inceasing.How do you explain this result. 

There are some writing errors,you should check again

Author Response

We thank the Reviewer for his/her comments and help to improve the quality of our manuscritp. Attached there is the point by point reply and manuscript modifications accordingly to Revier requests.

Reviewer 2 Report

interesting paper which follows previous results and established technology from the authors to expand the scope of OECT based viral infection monitoring

well designed and developed the paper lacks little in quality and care of presentation (say the lack of bar scale in Fig S1, or the erroneous data presentation in figure legends (i.e. a) vs b) in Fig 2, legend of Fig. S2, ...)

moreover it would be appropriate to improve comments about some key topics of the study:

#) how could be motivated the OECT sensitivity to "internal" cellular modification such as the B-CoV induced vacuoli formation ? - some vacuoli associated cellular modification of the electrolyte ? there is literature about ?

#) the very similar rates from 4 to 20 h in both "adhesion" and "suspension" tests would require some comments as it would be expected that a cell modification on population adhering on the channel should affect more strongly OECT responds than those in suspended cells

#) in figure 4 a) more comments about the ctrl vs 1/300 response in the time interval 20-50 h - some unexpected behavior

Author Response

We thank the Reviewer for his/her suggestions and contributions to our paper, allowing us to better explain and to improve the work. Attached there is the point-by-point reply to reviewer comments and consequent manuscript modifications
